# All-Solid-State Lithium Ion Batteries Using Self-Organized TiO_2_ Nanotubes Grown from Ti-6Al-4V Alloy

**DOI:** 10.3390/molecules25092121

**Published:** 2020-05-01

**Authors:** Vinsensia Ade Sugiawati, Florence Vacandio, Thierry Djenizian

**Affiliations:** 1Mines Saint-Etienne, Center of Microelectronics in Provence, Department of Flexible Electronics, F-13541 Gardanne, France; vinsensia.sugiawati@emse.fr; 2CNRS, Electrochemistry of Materials Research Group, Aix Marseille Université, MADIREL, UMR 7246, F-13397 Marseille CEDEX 20, France; florence.vacandio@univ-amu.fr; 3Center of Physical-Chemical Methods of Research and Analysis, Al-Farabi Kazakh National University, Tole bi str. 96A., Almaty 050000, Kazakhstan

**Keywords:** all-solid-state Li-ion batteries, TiO_2_ nanotubes, polymer electrolyte, anodization

## Abstract

All-solid-state batteries were fabricated by assembling a layer of self-organized TiO_2_ nanotubes grown on as anode, a thin-film of polymer as an electrolyte and separator, and a layer of composite LiFePO_4_ as a cathode. The synthesis of self-organized TiO_2_ NTs from Ti-6Al-4V alloy was carried out via one-step electrochemical anodization in a fluoride ethylene glycol containing electrolytes. The electrodeposition of the polymer electrolyte onto anatase TiO_2_ NTs was performed by cyclic voltammetry. The anodized Ti-6Al-4V alloys were characterized by scanning electron microscopy and X-ray diffraction. The electrochemical properties of the anodized Ti-6Al-4V alloys were investigated by cyclic voltammetry and chronopotentiometry techniques. The full-cell shows a high first-cycle Coulombic efficiency of 96.8% with a capacity retention of 97.4% after 50 cycles and delivers a stable discharge capacity of 63 μAh cm^−2^ μm^−1^ (119 mAh g^−1^) at a kinetic rate of C/10.

## 1. Introduction

Lithium-ion batteries (LIBs) have attracted great interest as excellent reversible energy storage devices due to their high energy density, low self-discharge, long cycle life, and several other benefits [1,2]. To gain the advanced storage performance, so far, most research has focused on innovative electrode materials, such as nanomaterials electrodes, which play a crucial role in recent technologies to reach high-performance devices [3,4,5]. Titanium dioxide (TiO_2_) has been considered as a good candidate as an anode material for LIBs due to its superior performances such as a low volume change of less than 4% during the reversible insertion of lithium ions, having a theoretical capacity of ca. 330 mAh g^−1^, good cycle life, low cost, and low toxicity [6]. Moreover, a high potential of 1.7 V versus Li/Li^+^ provides a good safety compared to commercialized graphite materials [7].

One-dimensional (1D) TiO_2_ materials have been developed as anodes for LIBs such as nanotubes, nanowires, nanorods, etc. [8]. Among the various nanostructured oxide materials, special attention has been directed to TiO_2_ nanotubes (TiO_2_ NTs) because they have been already explored for many applications such as solar cells [9,10,11], sensors [12,13], photocatalysis [14,15,16], and rechargeable batteries [17,18]. In addition, the TiO_2_ NTs materials have been extensively studied as anode material, thanks to the high surface-to-volume ratio leading to the enhanced electrochemical properties [19].

Several methods have been employed for the self-assembly of nanotubular arrays on titanium surfaces [20,21,22,23]. Among the approaches, the anodization technique is one of the most known and straightforward methods, which offers an extensive area of uniform nanotubes on Ti foil and allows the thickness of film oxide layer to be fine-tuned [13]. Recently, rechargeable all-solid-state batteries have become increasingly important to enable the miniaturization of electronic devices, the prevention of hazardous liquids, the deterioration of electrodes during cycling, and the absence of electrolyte leakage [7]. Planar thin-film solid-state Li-ion batteries have been rapidly emerged and extensively studied thorough vacuum deposition processes [24,25,26], but several potential drawbacks are evident, such as using highly reactive lithium; the use of highly reactive metallic lithium anode requires expensive packaging technology. In addition, pure lithium is extremely volatile and has a low melting point of approximately 180.5 °C, which is typically lower than that in the re-flow soldering process [27].

Many efforts have been made to improve the physical and electrochemical performance of the TiO_2_ nanotubes such as doping with foreign atoms (N, Nb, S, V, etc.) and carbon coating [28]. For example, Li et al. [29] prepared an N-doped TiO_2_ nanotubes/N-doped graphene composite via hydrothermal synthesis followed by a heat treatment in the presence of urea; the galvanostatic cycling test demonstrated a discharge capacity of 90 mAh g^−1^ at 5 A g^−1^. Kyeremateng et al. [30] reported that Tin-doped TiO_2_ nanotubes delivered higher reversible capacity compared to un-doped TiO_2_ nanotubes. In addition, Lopez et al. [31] prepared nanostructured electrodes based on the surface modification of TiO_2_ nanotubes with a low amount of Li_3_PO_4_, and their electrochemical performances showed good cycling stability. In the present work, we report the fabrication of an all-solid-state battery consisting of TiO_2_ NTs synthesized from ternary titanium alloy (Ti-6Al-4V, with 6 wt% aluminum and 4 wt% vanadium) alloy as an anode, a thin layer of polymer electrolyte, and an LiFePO_4_ layer as a cathode. Electrochemical tests show an improved performance of anodized Ti-6Al-V alloy after electropolymerization, exhibiting good discharge capacity, Coulombic efficiency, and capacity retention.

## 2. Results

### 2.1. Structural and Morphological Characterization

Figure 1 displays the XRD pattern of the TiO_2_ NTs before and after annealing treatment. As can be seen, some peaks located at 2θ = 35.6°, 38.7°, 40.6°, 53.3°, 63.6°, 71.1°, and 76.9° (Titanium, JCPDS 01-1197) can be attributed to Ti metal, and no visible diffraction peaks assigned to crystalline TiO_2_ are observed after anodization, indicating that the samples are amorphous [29,30]. The samples reveal sharp and strong peaks, which indicate that the as-annealed TiO_2_ NTs arrays have crystallized. The typical peaks observed at 2θ = 25.4° and 48.3° can be assigned to crystal phases of TiO_2_ (tetragonal, anatase, JCPDS 21-1272); the anatase peaks correspond to the planes (101) and (002), respectively. Furthermore, no peak of Al and V oxides are observed in the XRD patterns, which may be due to the very low percentage in the sample [32].

The water content plays a significant role in the dissolution rate, meaning that the increase in the water content leads to the faster dissolution of the oxide layer. Figure 2 shows the SEM micrographs of the anodized nanotubes grown on Ti-6Al-4V alloy in the ethylene glycol electrolyte with different water contents (20 wt% and 25 wt%). When the water content of 20% is employed, vertically aligned and uniformly opened TiO_2_ NTs are distributed over the alloy substrate with accessible pores on the surface. An inner diameter of the TiO_2_ NTs varies between 90 and 210 nm. Indeed, the TiO_2_ NTs structure does not collapse, implying that the nanotube arrays maintained better thermostability during annealing treatment at 450 °C. The length of the tubes of 1.25 µm has been estimated from the SEM cross-section view (see the inset Figure 2a). In contrast, at higher water content of 25 wt%, the surface shows some debris, and the appearance of the surface is less uniform (Figure 2b). This phenomenon is commonly found in polyphase alloys due to the presence of different phases, resulting in a preferential dissolution of less stable elements. The selective dissolution of heterogeneous alloy is reduced at a water content of 20 wt%, and the nanotube arrays formed uniformly for α and β phases.

Figure 2c illustrates the energy-dispersive spectroscopy (EDS) analysis for the elemental identification of the anodized Ti-6Al-4V alloy. The EDS spectrum of the sample indicates that Ti, O, Al, and V are present at the surface of anodized Ti-6Al-4V alloy. The strong percentage of O suggests that the oxidation of all elements occurred. In contrary to anodized pure Ti, F is not incorporated in oxides during the anodization process.

As reported in the literature [33,34], the growth reactions of the TiO_2_ NTs involve the field-assisted oxidation reaction of metal, field-assisted dissolution, and the chemical dissolution of the anodic layers. In this work, once a high potential of 60 V is applied to Ti alloys, an oxidation reaction occurs at the metal/metal oxide interface to generate Ti^4+^. The interaction of Ti^4+^ with O^2-^ and OH^-^ ions supplied by water molecules in the electrolyte produces a barrier layer of TiO_2_ and Ti(OH)_4_. Eventually, the Ti(OH)_4_ creates TiO_2_ and releases water via a condensation reaction. Meanwhile, the hydrogen evolution takes place to form hydrogen ions and electrons at the Pt counter electrode surfaces [35]. The chemical reactions involved during this process are represented in the following Equations (1)–(5):Ti → Ti^4+^ + 4e^−^(1)
Ti^4+^ + 2O^2−^ → TiO_2_(2)
Ti^4+^ + 4OH^−^ → Ti(OH)_4_(3)
Ti(OH)_4_ → TiO_2_ + 2H_2_O(4)
4H^+^ + 4e^−^ → 2H_2_.(5)

An overall reaction of the Ti reacts with water generating the oxide layer and hydrogen gas is written as follows:Ti + 2H_2_O → TiO_2_ + 2H_2_.(6)

The formation of TiO_2_ nanotubes is generally associated with the presence of fluoride ions. At the high applied potential, the formed compact oxide layer, Ti(OH)_4_, and Ti^4+^ ions are chemically reacting with F^−^ to form water-soluble fluoride complexes [TiF_6_]^2−^. The selective dissolution of aluminium oxide (Al_2_O_3_) and vanadium oxides (e.g., VO_2_) might be occurred simultaneously at this stage. As a result, the balance of the metal oxidation and chemical dissolution of the oxide layer leads to the growth nanotubular structures. The dominant reaction mechanisms are proposed in Equations (7)–(9) as follows:TiO_2_ + 6F^−^ + 4H^+^ → [TiF_6_]^2−^ + H_2_O(7)
Ti(OH)_4_ + 6F^−^ → [TiF_6_]^2−^ + 4OH^−^(8)
Ti^4+^ + 6F^−^ → [TiF_6_]^2−^.(9)

### 2.2. Electropolymerization of PMMA–PEG Polymer Electrolyte

In the present study, the electrochemical deposition of the polymer electrolyte into TiO_2_ NTs is achieved by the cyclic voltammetry (CV) technique. The nanotubes synthesized in ethylene glycol electrolyte containing 20 wt% water are used for the electropolymerization of the polymer electrolyte into the nanotubes. The starting monomer is methyl ether methacrylate poly (ethylene glycol) (MMA-PEG, Mw: 500). The upper potential limit was set according to the open circuit voltage of the system, which is around −0.35 V versus Ag/AgCl (3M KCl), and the lower limit was set to −1V versus Ag/AgCl (3M KCl) [36]. The starting monomers are electropolymerized into short chains of PMA-PEG, where PMA is used to enhance the mechanical properties and PEG is responsible for the ionic conductivity of the polymer electrolyte. The reaction mechanism of MMA-PEG electropolymerization into TiO_2_ NTs is based on the formation of intermediate hydrogen-free radicals [37]. For cathodic applied potentials, the reduction of H^+^-producing H_2_ is accompanied by the formation of hydrogen-free radicals that can react with the monomers, leading to the polymerization of methyl methacrylate (PMMA). Figure 3 shows the cyclic voltammogram of the polymer-coated TiO_2_ NTs for 25 cycles. Clearly, the absolute value of the cathodic current at −1 V versus Ag/AgCl (3M KCl) drops as the increment of cycles. The reason for the obvious fading can be explained by the successive growth of thin insulating polymer layers on the walls of TiO_2_ NTs.

The electrodeposition of the polymer electrolyte into the TiO_2_ NTs layer was confirmed by morphological analysis. Figure 4a–f shows the SEM images of polymer-coated TiO_2_ NTs after a different number of CV cycles such as 5, 10, and 25 cycles, confirming the conformal deposition of the polymer electrolyte. After 25 cycles, the nanotubes are filled and almost covered by a thin polymer film, following the surface rugosity of the surface. The polymer electrolyte must be conformally deposited onto the electrodes and form a continuous layer without any pinholes in order to be used as separator to prevent the short-circuiting of the electrodes.

### 2.3. Electrochemical Performance of Polymer-Coated TiO_2_ NTs

To gain deeper insight into the positive influence of the electropolymerization approach, the bare TiO_2_ NTs and the polymer-coated TiO_2_ NTs were investigated as anode materials for Li-ion batteries. Herein, the synthesized TiO_2_ NTs with 20 wt% H_2_O has been used to study the charging and discharging mechanisms during the electrochemical cycling experiments. The samples were tested at scanning rates of 0.5 mV s^–1^ in the potential range between 1 and 3 V versus Li/Li^+^ at room temperature, as displayed in Figure 5a,b. The distinctive anodic and cathodic peaks represent the redox reactions of the electrodes with Li^+^, which is responsible for the electrode phase transformations. The redox peaks are well-defined and distinguishable (Figure 5a). The Li^+^ insertion (cathodic peak) and extraction (anodic peak) occur at the potential of 1.68 and 2.06 V versus Li/Li^+^, respectively. The slightly decrease of the current densities in the absolute cathodic and anodic peak indicates the small discharge capacity fading upon cycling. However, after first cycle, well-overlapped redox peaks are visible, indicating a good electrochemical stability of the electrode. An additional peak pair at a potential of approximately 2.62 V versus Li/Li^+^ with a low current density is also appeared due to the presence of an electrochemically active VO_2_ phase, but the VO_2_ phase does not significantly contribute to the storage performance of the electrode [31]. The similar behavior is observed for the bare TiO_2_ NTs, showing the anodic peak at 2.08 V versus Li/Li^+^ and cathodic peak at 1.63 V versus Li/Li^+^ (Figure 5b). The CV of the bare TiO_2_ NTs clearly indicates a lower electrochemical activity and lithium ions storage compared to that of polymer-coated TiO_2_ NTs, as shown by a smaller sweeping area and a lower oxidation and reduction peak current density. Moreover, the peak separation of the polymer-coated TiO_2_ NTs is substantially lower than that of bare TiO_2_ NTs, suggesting a lower polarization.

Figure 5c,d depicts the galvanostatic charge–discharge profiles of polymer-coated TiO_2_ NTs. The profiles display flat plateaus of charge at 1.85 V versus Li/Li^+^ and discharge 1.76 V versus Li/Li^+^, which are typical profiles for anatase TiO_2_ NTs as obtained from XRD results. The first charge and discharge capacities for polymer-coated TiO_2_ NTs are 74.9 µAh cm^−2^ µm^−1^ (176.8 mAh g^−1^) and 139.2 µAh cm^−2^ (328.5 mAh g^−1^) µm^−1^, respectively, corresponding to a Coulombic efficiency (CE) of 53.78%. Meanwhile, the bare TiO_2_ NTs shows flat plateaus of charge at 1.86 V versus Li/Li^+^ and discharge 1.74 V versus Li/Li^+^, a delivering charge capacity of 59 µAh cm^−2^ µm^−1^ (139.2 mAh g^−1^) and a discharge capacity of 132.5 µAh cm^−2^ µm^−1^ (312.7 mAh g^−1^) with a significantly lower CE of 44.5% compared to that of polymer-coated TiO_2_ NTs. The specific discharge capacity of the polymer-coated TiO_2_ NTs is higher compared to some previous studies. For instance, Fasakin et al. [38] reported that an anatase TiO_2_ NTs prepared from pristine anatase TiO_2_ nanoparticles via a low temperature modified stirring-hydrothermal technique delivered a capacity of 160 mAh g^−1^ at a specific current of 36 mA g^−1^. Auer et al. [39] studied the TiO_2_ NTs synthesized by a two-step anodization process, and the anatase TiO_2_ NTs exhibited a reversible capacity of 185 mAh g^−1^ (Li_0.55_TiO_2_) at a slow C-rate (C/20). In addition, Savva et al. [40] synthesized ordered TiO_2_ NTs via electrochemical anodization and subsequently annealed under N_2_ and water-vapor (WV) atmospheres. The obtained capacities for the N_2_ and WV-treated TiO_2_ NTs were 231.9 mAh g^−1^ and 230.6 mAh g^−1^ at C/20, respectively.

The cells were cycled at multiple C-rates as presented in Figure 5e. A bare TiO_2_ NTs layer exhibits a stable capacity of 41.8 µAh cm^−2^ µm^−1^ (98.6 mAh g^−1^) at C/5, 25.9 µAh cm^−2^ µm^−1^ (61.1 mAh g^−1^) at C/2, and 17.2 µAh cm^−2^ µm^−1^ (40.6 mAh g^−1^) at 1C. Obviously, there is a significant capacity increase for the polymer-coated TiO_2_ NTs compared to the bare TiO_2_ NTs for all C-rates. Polymer-coated TiO_2_ NTs layers deliver a stable capacity of 53.6 µAh cm^−2^ µm^−1^ (126.5 mAh g^−1^) at C/5, 36.9 µAh cm^−2^ µm^−1^ (87.1 mAh g^−1^) at C/2, and 27.4 µAh cm^−2^ µm^−1^ (64.7 mAh g^−1^) at 1C. The capacities can be recovered at C/10 for both samples. Herein, we note a large irreversible capacity at the first initial cycles for both samples, which may originate from the side reaction between the electrodes and Li^+^, as well as the presence of residual water at the surface of TiO_2_ NTs after electropolymerization. This phenomenon is in agreement with the previous reports [41]; the polymerization of MMA monomer during the electrochemical cycling test could occur, and the increasing viscosity of the electrolyte might lead to the decrement of the ionic conductivity. The enhanced capacity and good rate capability of the polymer-coated TiO_2_ NTs can be ascribed to the lower interfacial resistance due to the better interfacial contact between the anode and polymer electrolyte; hence, it also enhances the diffusion of Li ions. Indeed, the Coulombic efficiency of the electrode is significantly improved close to 100% upon cycling (Figure 5f). It should be noted that the increment in the capacity is obtained for the polymer-coated TiO_2_ NTs for 25 cycles CV due to the enhancement of the electrode–electrolyte interface between the TiO_2_ NTs and the polymer electrolyte. Additionally, the electropolymerization process helps in exploiting the full active surface area of the nanotubes and hence obtaining the contribution from the whole electrode surface.

### 2.4. Fabrication and Characterization of All-Solid-State Batteries Based on Anodized TI-6Al-4V Alloy and LiFePO_4_

Herein, a LiFePO_4_ cathode is paired with polymer-coated TiO_2_ NTs in the full-cell system due to its stability, low cost, and environmental friendliness [42]. Compared to many other cathode compositions, LiFePO_4_ has a lower voltage of 3.45 V versus Li/Li^+^. However, LiFePO_4_ shows flat charge–discharge curves during the two-phase Li insertion–extraction process and excellent cycling stability due to its unique ordered olivine structure [43]. The electrochemical reactions of LiFePO_4_ with Li using a polymer electrolyte have been studied by cyclic voltammetry. Figure 6a shows the CV curves obtained in the potential range of 2–4.2 V versus Li/Li^+^ at a scan rate of 0.5 mV s^−1^. The oxidation peak is visible at 3.75 V versus Li/Li^+^, while the reduction peak appears at 3.2 V versus Li/Li^+^. These redox peaks correspond to the reversible insertion/extraction of Li ions in the LiFePO_4_ electrode. Galvanostatic tests revealed that the reversibility of LiFePO_4_ cathode is reflected in the voltage profiles with plateaus at the corresponding voltages (Figure 6b). The charge and discharge plateaus located at 3.57 V versus Li/Li^+^ and 3.31 V versus Li/Li^+^, respectively, originate from the redox process between Fe^3+^ and Fe^2+^. The initial reversible capacity is 118 mAh g^−1^ at a rate of C/10, which is followed by a capacity of 117.8 mAh g^−1^ in the second cycle; afterwards, the voltage profiles and capacity are retained. A slight decrease in the capacity could be attributed to a lower ionic conductivity of the MMA-PEG polymer electrolyte compared to that of organic liquid electrolyte, resulting in an internal resistance (IR) drop at the anode–electrolyte interface.

The promising performances of the polymer-coated TiO_2_ NTs suggested it as a suitable anode material for application in full all-solid-state Li-ion batteries using the LiFePO_4_ cathode material, as schematically depicted in Figure 7a. Note that for this battery, the polymer electrolyte is deposited via electropolymerization reaction for 25 cycles CV. The polymer thin film deposited atop the TiO_2_ NTs during the electropolymerization process serves as a continuous layer with an additional drop-cast polymer layer, with the total thickness of the polymer electrolyte layer being approximately 200 µm. As seen in Figure 7b, the electrochemical behavior of TiO_2_ NTs/polymer/LiFePO_4_ full-cell by performing the cyclic voltammetry (CV) test is investigated. The CV curves of the full-cell recorded at the scanning rate of 0.5 mV s^−1^ exhibit well-defined redox peaks. The presence of the oxidation peaks at 1.93 V is attributed to the extraction of Li^+^ from the cathode and their subsequent insertion into the anode material. By contrast, reduction peaks at 1.21 V are assigned to the extraction of Li^+^ from the anode and their simultaneous insertion into the cathode according to the reactions given in Equations (10)–(12).
Cathode: LiFePO_4_→ xLi^+^ + xe^−^ + Li_(1−x)_ FePO_4_(10)
Anode: TiO_2_ + xLi + xe^−^ → Li_x_TiO_2_(11)
Overall reaction: LiFePO_4_ + TiO_2_ ⇌ LixTiO_2_ + Li _(1−x)_ FePO_4_(12)

Since the limiting reactant material of the thin-film battery is TiO_2_ NTs, the capacity values are reported versus the mass and the surface of the anode. Figure 7c shows the galvanostatic charge/discharge profile of the battery consisting of a TiO_2_ NTs/Polymer/LiFePO_4_ full-cell in the potential window of 0.5–2.8 V cycled at the C/10 rate. At the first cycle, the initial charge and discharge capacities recorded at the C/10 rate are 66 μAh cm^−2^ μm^−1^ (155.8 mAh g^−1^) and 63.9 μAh cm^−2^ μm^−1^ (150.8 mAh g^−1^), resulting in a high Coulombic efficiency (CE) of 96.8% and high capacity retention of 97.4% after 50 cycles at the C/10 rate. The discharge plateau at approximately 1.75  V and a charge plateau at approximately 1.50 V almost match with the two intensive peaks in the CV curves. By taking the middle point of two plateaus, we noted that the operating cell voltage of the TiO_2_ NTs/Polymer/LiFePO_4_ full-cell is approximately 1.6 V. An additional plateau observed at 0.75 V could be attributed to the insertion of Li^+^ from LiFePO_4_ to the VO_2_ structure. This result is in agreement with the galvanostatic cycling, resulting in half-cell TiO_2_ NTs on Ti-6Al-4V alloy where the plateau at 2.62 V corresponds to the VO_2_ phase.

Figure 7d,e shows the rate performance of a TiO_2_ NTs/polymer/LiFePO_4_ full-cell recorded at multi-C rates for 50 cycles. The battery delivers capacity values of 63 μAh cm^−2^ μm^−1^ (119 mAh g^−1^) at C/10, 51 μAh cm^−2^ μm^−1^ (96.4 mAh g^−1^) at C/5, and 23 μAh cm^−2^ μm^−1^ (43.5 mAh g^−1^) at C/2, which are well recovered after the fast cycling rate (C/2). The high capacities for the TiO_2_ NTs/polymer/LiFePO_4_ full-cell are attributed to the enhanced surface area between the nanotubes and the gel polymer electrolyte, hence providing a robust and a high quality electrode–electrolyte interface for long charge–discharge cycles^36^. The better electrode–electrolyte contact established between the electrode and polymer electrolyte drastically facilitates the insertion and diffusion of Li^+^. Furthermore, calculated using a working voltage of approximately 1.6 V and taking into account the reversible areal capacity of the polymer-coated TiO_2_ NTs (69 μAh cm^−2^ μm^−1^), an areal energy density of 110.4 μWh cm^−2^ μm^−1^ and an areal power density of 11.04 μW cm^−2^ μm^−1^ at C/10 rate can be achieved.

## 3. Materials and Methods

### 3.1. Synthesis of Self-Organized TiO_2_ Nanotubes Grown on Ti–6Al–4V Alloy

The Ti–6Al–4V alloy (0.1 mm thick, 25% tolerance, Goodfellow) was cut into 1.2 cm × 1.2 cm and ultrasonically cleaned in acetone, 2-propanol, and methanol for 10 min each. Then, the treated foils were rinsed with ultrapure water and dried in compressed air. At room temperature, the anodization was performed in a two-electrode electrochemical cell with Ti-6Al-4V alloy as the working electrode and Pt foil as the counter electrode in ethylene glycol electrolyte containing 0.3 wt% ammonium fluoride (NH_4_F) with two different water contents (20 and 25 wt%). All anodization experiments were carried out under a constant voltage of 60 V using a generator (ISO-TECH IPS-603) for 3 h [32]. After the anodization, the samples were removed from the electrochemical cell and they were dried in compressed air. The as-prepared samples were annealed at 450 °C for 3 h with a heating rate of 5 °C/min.

### 3.2. Electropolymerization of MMA-PEG on TiO_2_ NTs

The electropolymerization was performed by the cyclic voltammetry (CV) technique in a three-electrode configuration consisting of the layer of TiO_2_ NTs as the working electrode, a Pt foil as the counter electrode, and an Ag/AgCl (3M KCl) as the electrode. An aqueous solution of lithium bis(trifluoromethanesulfonyl)imide (LiTFSI) and MMA-PEG monomer was mixed and stirred for 3 h; then, the solution was introduced into the cell. Prior to the electropolymerization, the solution was purged with N_2_ for 10 min to eliminate dissolved oxygen. Cyclic voltammetry tests were carried out in a cathodic current with a potential range of −0.35 to −1 V versus Ag/AgCl (3M KCl) at the scan rate of 10 mV s^−1^ for 5, 10, and 25 cycles. After the CVs, the sample was removed from the cell without rinsing and dried in the BUCHI vacuum dryer at 70 °C overnight to evaporate the residual water.

### 3.3. Preparation of Electrodes

For the TiO_2_ NTs anode, there was no need to further mix with any additive due to their self-supported structure. For the positive electrode materials, the LiFePO_4_ powder was synthesized as previously reported [41]. Each material was mixed with carbon black (Super P) and polyvinylidene fluoride (PVDF) at the ratio of 80:10:10. *N*-methyl-2-pyrrolidone (NMP) was added in the powder mixture to obtain a paste. The paste was spread on an Al disk (diameter of 8 mm) that was used as the current collector. The electrode was dried at 110 °C for 8 h.

### 3.4. Fabrication of the All-Solid-State Batteries

All-solid-state batteries were assembled using bare and polymer-coated TiO_2_ NTs and LiFePO_4_. After drying the polymer-coated TiO_2_ NTs electrode at 70 °C for 24 h, 7 µL of the MMA-PEG carrying LiTFSI salt was drop-casted at the polymer-coated TiO_2_ NTs surface. The sample was dried again under vacuum at 70 °C for 24 h to obtain a homogeneous polymer thin film. The cells consisting of a polymer-coated TiO_2_ NTs anode and an LiFePO_4_ cathode were assembled in an argon-filled glove box (MBraun, Munich, Germany) with <0.5 ppm H_2_O and <0.5 ppm O_2_ atmosphere. The LiFePO_4_ composite cathode on the Al disk was pressed to polymer-coated TiO_2_ NTs and a polymer electrolyte dropcast. Two wires were connected to the backside of each current collector using silver conductive paste and wires.

### 3.5. Characterizations and Measurements

The surface morphology of the self-organized TiO_2_ NTs was examined using a field-emission scanning electron microscope (FE-SEM, Ultra-55, Carl Zeiss, Oberkochen, Germany). The purity of TiO_2_ NTs was examined by X-ray diffraction (XRD) using a Diffractometer D5000 (Siemens, Munich, Germany) with CuK_α1_ (λ= 1.5406 Å) radiation. Then, they were analyzed by comparing with the JCDS-ICDD database (Joint Committee on Powder Diffraction Standards—International Center for Diffraction Data, Newtown Square, PA, USA) to check the purity of the samples.

For the Swagelok test cells employing the gel electrolyte, the separators were prepared by soaking each circular sheet (diameter of 10 mm) of the Whatman glass microfiber with 80 µL of a mixture of 10 mL of 0.5 M LiTFSI with 2.5 g MMA-PEG. The separators were dried overnight at 70 °C in the BUCHI vacuum dryer. For the electrochemical performance tests, bare TiO_2_ NTs and polymer-coated TiO_2_ NTs having a surface area of 0.50 cm^2^ were used as electrodes without the use of any binders and conductive additives, assembling using standard two-electrode Swagelok cells. All cells assembly were conducted in a glove box that was filled with high-purity argon atmosphere in which the moisture and oxygen contents were less than 0.5 ppm. Cyclic voltammetry tests were carried out using a VMP3 (Bio Logic, Seyssinet-Pariset, France) in the potential window of 1–3 V versus Li/Li^+^ with a scan rate of 0.5 mV s^−1^ for the bare TiO_2_ NTs and polymer-coated TiO_2_ NTs. Galvanostatic charge–discharge cycles were performed with a VMP3 (Bio Logic) in the potential window between 1 and 3 V versus Li/Li^+^. The Swagelok test cells were assembled in an Argon-filled glove box. In the half-cell system, TiO_2_ NTs electrodes were assembled against metallic Li foil using the gel electrolyte of LiTFSI and MMA-PEG. The Li foil was cut in a circular shape with the diameter of 9 mm, and two circular sheets (10 mm in diameter) of the gel electrolyte in the Whatman paper were used as the separator. For the LiFePO_4_ electrodes, cyclic voltammetry tests were carried out in the potential window of 2–4.2 V versus Li/Li^+^ with a scan rate of 0.5 mV s^−1^ and galvanostatic charge–discharge cycles were performed in the potential window between 2 and 4.2 V versus Li/Li^+^.

Full-cells consisting of the polymer-coated TiO_2_ NTs/polymer/LiFePO_4_ were cycled at different C-rates varying from C/10 to C/2 in the potential window of 0.5–2.8 V. The charge and discharge rates of a battery are expressed by the C-rate; it means a battery is charged and discharged relative to its maximum capacity. The rate *n*C means the current will charge and discharge the full capacity in 1/*n* hour.

## 4. Conclusions

In the present work, we showed that the water content in the electrolyte significantly affects the morphology of self-organized TiO_2_ nanotubes obtained by anodization of Ti-6Al-4V alloy. A vertically aligned and uniformly opened TiO_2_ NTs are distributed over the alloy substrate at 20 wt% H_2_O, and by adding 5 wt% H_2_O, the nanotubes have become less uniform. The electropolymerization of the MMA-PEG into TiO_2_ nanotubes was carried out at different CV cycles (5, 10, and 25 cycles). After 25 cycles, the nanotubes are almost covered by a thin polymer film following the nanotubes surface rugosity. As a result, there is a significant increase in the capacity of the polymer-coated TiO_2_ NTs compared to the bare TiO_2_ NTs in all C-rates due to the improvement of the electrode-electrolyte interface. Therefore, polymer-coated TiO_2_ NTs layers deliver higher stable capacity of 53.6 µAh cm^−2^ µm^−1^ (126.5 mAh g^−1^) at C/5, 36.9 µAh cm^−2^ µm^−1^ (87.1 mAh g^−1^) at C/2, and 27.4 µAh cm^−2^ µm^−1^ (64.7 mAh g^−1^) at 1C compared to that of bare TiO_2_ NTs. The full-cell battery was composed of a polymer-coated TiO_2_ NTs as an anode, a thin layer of MMA-PEG polymer electrolyte, and a composite LiFePO_4_ cathode shows an operating voltage of 1.6 V with good electrochemical performance, not only a stable discharge capacity of 62.2 μAh cm^−2^ μm^−1^ (117.6 mAh g^−1^) after 50 cycles versus anode at C/10 but also a high capacity retention of 97.35% and a Coulombic efficiency of 96.78% beyond 50 cycles.

## Figures and Tables

**Figure 1 molecules-25-02121-f001:**
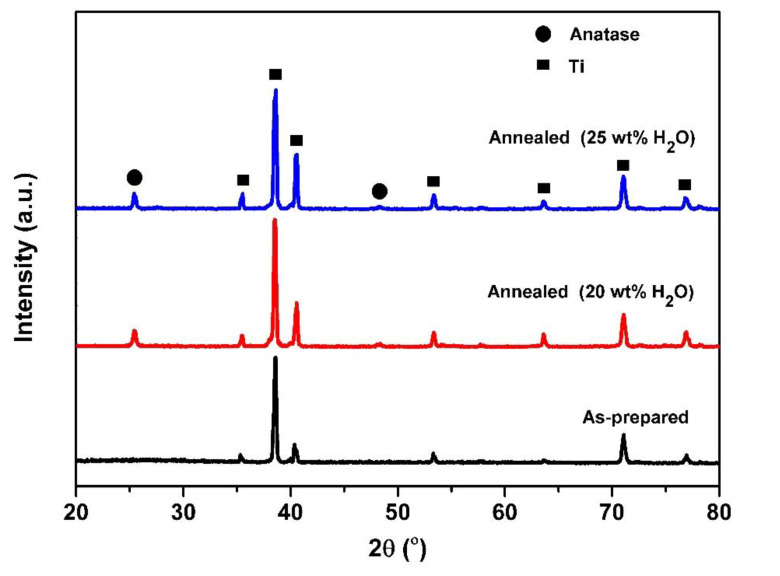
XRD patterns of TiO_2_ nanotubes (TiO_2_ NTs) obtained from anodization of Ti-6Al-4V alloy (before and after annealing treatment) at different water contents: 20 wt% and 25 wt% H_2_O.

**Figure 2 molecules-25-02121-f002:**
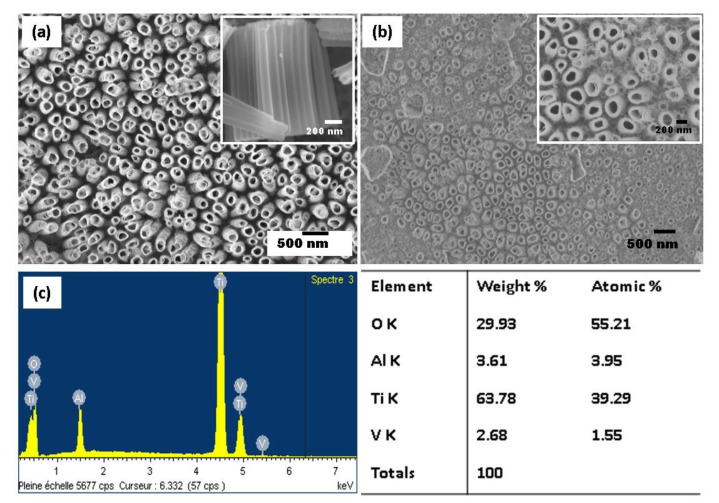
SEM images of TiO_2_ NTs obtained from the anodization of Ti-6Al-4V alloy in fluoride ethylene glycol electrolyte with different water contents: 20 wt% H_2_O (top view in the inset) (**a**) and 25 wt% H_2_O (**b**), and the energy-dispersive spectroscopy (EDS) elemental spectrum of the anodized Ti-6Al-4V alloy (**c**).

**Figure 3 molecules-25-02121-f003:**
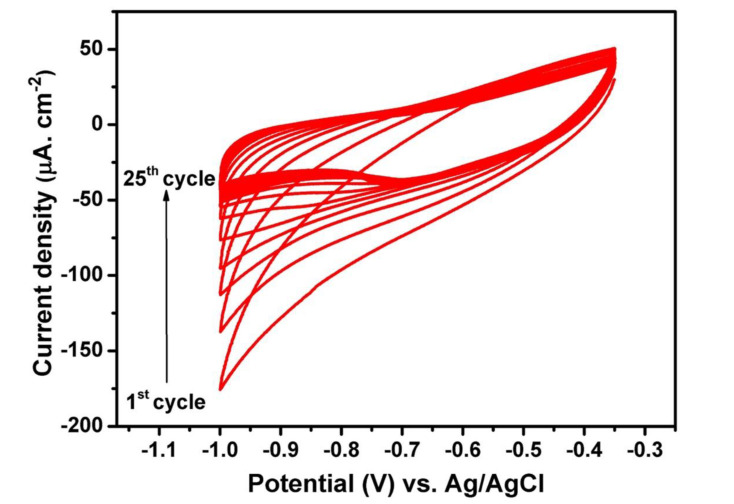
Cyclic voltammograms of TiO_2_ NTs electrode in 0.5 M LiTFSI + 0.5M methyl ether methacrylate poly (ethylene glycol) (MMA-PEG)500. The curves were recorded in the potential window of −0.35 to −1 V vs. Ag/AgCl (3M KCl) at the scan rate of 10 mV s^−1^.

**Figure 4 molecules-25-02121-f004:**
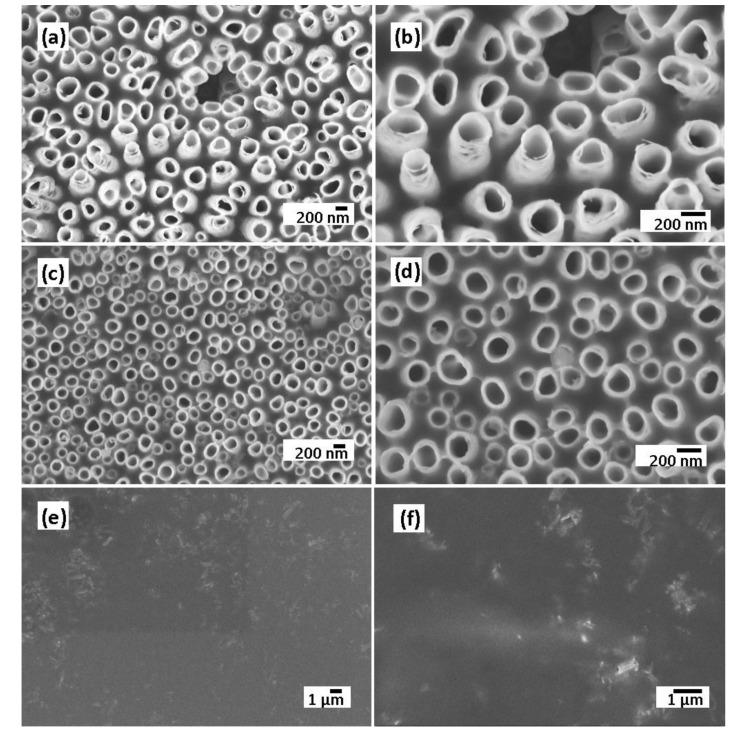
SEM images of TiO_2_ NTs electrode at low and high magnifications after 5 cycles (**a**,**b**), 10 cycles (**c**,**d**), and 25 cycles of CV (**e**,**f**).

**Figure 5 molecules-25-02121-f005:**
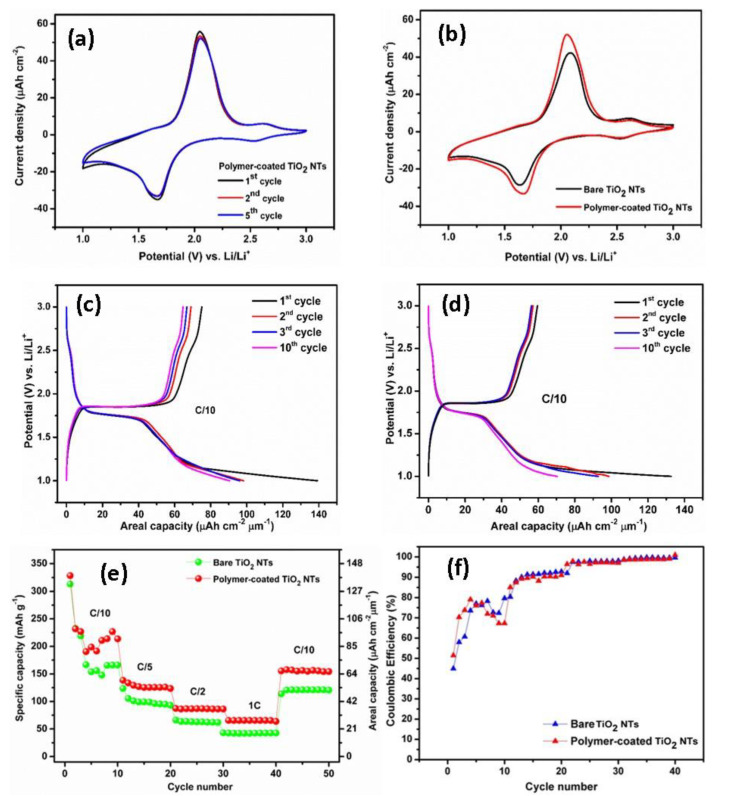
Cyclic voltammogram of the polymer-coated TiO_2_ NTs (**a**). Comparison of cyclic voltammograms between bare TiO_2_ NTs and polymer-coated TiO_2_ NTs (**b**). Typical galvanostatic charge–discharge profiles of the polymer-coated TiO_2_ NTs (**c**) and bare TiO_2_ NTs (**d**). Rate performance of the bare and polymer-coated TiO_2_ NTs (**e**) and Coulombic efficiencies of the bare TiO_2_ NTs and polymer-coated TiO_2_ NTs at C/10 to 1C rate (**f**).

**Figure 6 molecules-25-02121-f006:**
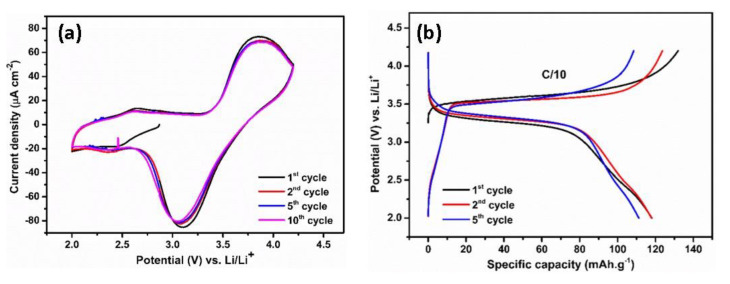
Cyclic voltammogram (**a**) and typical galvanostatic charge/discharge profiles (**b**) of LiFePO_4_ at C/10 rate in the potential range of 2–4.2 V vs. Li/Li^+^.

**Figure 7 molecules-25-02121-f007:**
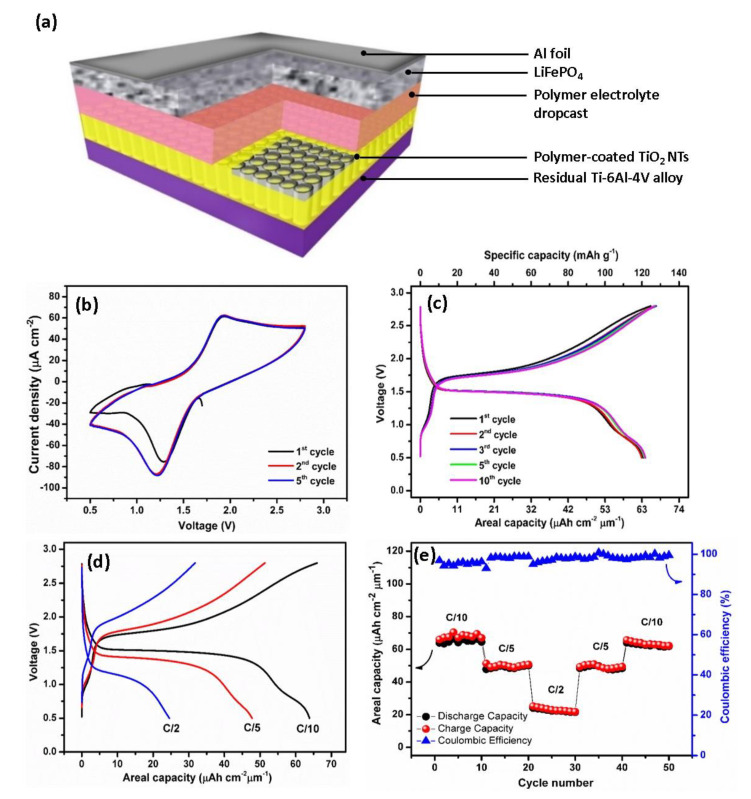
Schematic representation of the all-solid-state battery (**a**). Cyclic voltammograms of TiO_2_ NTs/polymer/LiFePO_4_ full-cell at a scan rate of 0.5 mV s^−1^ in the potential range of 0.5–2.8 V (**b**). Galvanostatic charge/discharge profiles of TiO_2_ NTs/polymer/LiFePO_4_ full-cell at the C/10 rate in the potential range of 0.5–2.8 V (**c**). Charge–discharge profiles of TiO_2_ NTs/polymer/LiFePO_4_ full-cell at various current densities from C/10 to C/2 (**d**). Rate capability of TiO_2_ NTs/polymer/LiFePO_4_ full-cell (**e**).

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
