# Peer review of "All-Solid-State Lithium Ion Batteries Using Self-Organized TiO2 Nanotubes Grown from Ti-6Al-4V Alloy"

_molecules, 2020, doi:10.3390/molecules25092121_

Round 1

Reviewer 1 Report

In this manuscript, the authors propose a facile route to prepare self-organized TiO2 nanotubes grown on Ti-6Al-4V alloy, the structural characterizations and electrochemical performances have been carried out and measured concerning lithiation/delithiation behaviors of the electropolymerized TiO2 nanotubes anodes. It is interesting and has shown better potential for applications. However, it currently suffers from a few issues need to be properly addressed, prior to acceptance for publication. The comments are provided as follows:

  1. The acronym of “TiO2nts” should be defined in the first appearance in text, for example, in the abstract. The format of “TiO2 NTs” seems more acceptable by readers.
  2. In the inset of Fig. 2(b), the scale bar should be highly magnified. In Fig. 4, the magnifications should be the same among relevant pictures in high quality.  
  3. In section 2.3, lines 162-163, the author says “The slightly decrease in the absolute cathodic and anodic current…….”, however, the result shows an increased tendency of current density at redox peaks with cycling, could authors explain the reason or correct?
  4. Performance comparison with other kind of one-dimensional TiO2 materials is suggested, thus to well highlight the improved performances of these TiO2 NTs.
  5. In Fig. 6, the test potential range is 0.5-2.8 V (also in the section 2.4, line 214), however, it disagrees with the experimental result, a further check is necessary.
  6. Some grammatical errors and unclear expression are found in the text, careful revisions are requested.

Author Response

Point 1: The acronym of “TiO2nts” should be defined in the first appearance in text, for example, in the abstract. The format of “TiO2 NTs” seems more acceptable by readers

Response 1: We absolutely agree with the reviewer. The acronym of “TiO2nts” has been replaced by “TiO2 NTs”

Point 2: In the inset of Fig. 2(b), the scale bar should be highly magnified. In Fig. 4, the magnifications should be the same among relevant pictures in high quality.  

Response 2: This is a good point. The SEM images have been changed, the scale bar has been magnified for fig. 2b.

Point 3: In section 2.3, lines 162-163, the author says “The slightly decrease in the absolute cathodic and anodic current…….”, however, the result shows an increased tendency of current density at redox peaks with cycling, could authors explain the reason or correct?

Response 3: Thank you for this valuable comment. The reviewer is right, we have corrected the figure.

Point 4: Performance comparison with other kind of one-dimensional TiO2 materials is suggested, thus to well highlight the improved performances of these TiO2 NTs.

Response 4: Thank you for this suggestion. We also provided the specific capacity in order to its electrochemical performance with the other materials. We have added the performance comparison as shown in the text with yellow background.

"The specific discharge capacity of the polymer-coated TiO2 NTs is higher compared to some previous studies. For instance, Fasakin et al. [38] reported that an anatase TiO2 NTs prepared from pristine anatase TiO2 nanoparticles via a low temperature modified stirring-hydrothermal technique delivered a capacity of 160 mAh g-1 at a specific current of 36 mA g-1. Auer et al. [39] studied the TiO2 NTs synthesized by a two-step anodization process and the anatase TiO2 NTs exhibited a reversible capacity of 185 mAh g-1 (Li0.55TiO2) at slow C-rate (C/20). Also, Savva et al. [40] synthesized ordered TiO2 NTs via electrochemical anodization and subsequently annealed under N2 and water-vapor (WV) atmospheres. The obtained capacities for the N2 and WV-treated TiO2 NTs were 231.9 mAh g-1 and 230.6 mAh g-1 at C/20, respectively"

Point 5: In Fig. 6, the test potential range is 0.5-2.8 V (also in the section 2.4, line 214), however, it disagrees with the experimental result, a further check is necessary

Response 5: Thank you for this valuable comment. We have changed the potential range for the LiFePO4. Actually, the potential range for the half-cell: TiO2 NTs (1-3 V), LiFePO4 (2-4.2) and TiO2 NTs vs. LiFePO4 full cell (0.5-2.8 V).

Point 6: Some grammatical errors and unclear expression are found in the text, careful revisions are requested.

The revision has been done.

Again, we appreciate all of your insightful comments. Thank you for taking the time and energy to help us improve the paper.

Reviewer 2 Report

This manuscript entitled in “Improved Performance of All-Solid-State Lithium Ion Batteries Using Electropolymerized TiO2 Nanotubes Grown on Ti-6Al-4V Alloy” describes the electrochemical characterization of 3D-TiO2 Nanotubes/Li-ion conductive MMA-PEG polymer electrolyte composite anodes by using various cell architectures. This article has much overlap with the contents of Reference 38, and no novelty is recognized in most of main contents. Although the application of the same composite electrode structures to a solid-state battery system is seemed to be progressed as comparing to Ref. 38, the title and conclusion described in this paper does not match the main contents. The reviewer strongly suggests rewriting the appropriate title and conclusion of this paper prior to resubmit elsewhere. Furthermore, the reviewer thinks that there are many lack of results and explanations for understanding the impact of direct polymerization of MMA-PEG polymer form the TiO2 nanotube electrodes on their kinetical parameters in the all solid state battery which is proposed by the authors. The reviewer really hates to tell the author, but I cannot recommend that this manuscript can be accepted for publication in “Molecules”.

Author Response

We thank the reviewer for the comments. Our previous work, mentioned as ref 38, the title of the paper is high energy and power density TiO2 nanotube electrodes for single and complete lithium-ion batteries.

  1. The TiO2 NTs was grown from Ti foil, in this work from Ti alloy with different electrolyte composition.
  2. The previous work also investigated the amorphous TiO2 NTs, not anatase as we can see from the galvanostatic cycling, there are no plateaus, indicating the amorphous structure of TiO2 NTs.
  3. In the previous work, we used whatman paper as separator to assemble the full-cell with the thickness of the separators around 520 µm and the Swagelok cells (not wire connections).

It is thick enough for the all-solid-state as we know the lower ionic conductivity of polymer compared to the common organic liquid electrolyte. The thicker separator makes slower lithium diffusion during the insertion/extraction of Li-ions, so we tried to remove the thicker separator by increasing the no of cycles (EP 25 cycles + drop cast), in this report the PMMA-PEG acts as polymer electrolyte and separator (no whatman paper and LiPF6 (EC:DEC)), etc.

We submitted this manuscript as “Communication” for publication in “Molecules”

Again, we appreciate all of your insightful comments. Thank you for taking the time and energy to help us improve the paper.

Reviewer 3 Report

The article is interesting, but in my opinion EDS and/or XPS results are missing.

Author Response

The authors would like to thank the editor and the reviewers for their precious time and
comments which were useful inputs for improving our manuscript. We have carefully addressed all the comments. The corresponding changes and refinements made in the revised paper are summarized in our response below.

Point 1: The article is interesting, but in my opinion EDS and/or XPS results are missing.

Response 1: Thanks for raising this important point. The EDS analysis has been added for the elemental identification as shown in the text with yellow background.

"Figure 2c illustrates the energy-dispersive spectroscopy (EDS) analysis for the elemental identification of the anodized Ti-6Al-4V alloy. The EDS spectrum of the sample indicates that Ti, O, Al, and V are present at the surface of anodized Ti-6Al-4V alloy. The strong percentage of O suggests that oxidation of all elements occurred. In contrary to anodized of pure Ti, F is not incorporated in oxides during the anodization process."

Please, note that because of COVID-19, access to laboratories are strictly prohibited and XPS analysis cannot be planned for the next months.

Again, we appreciate all of your insightful comments. Thank you for taking the time and energy to help us improve the paper.

Reviewer 4 Report

This is a well written paper with interesting results reported. I recommend the publication after minor changes.

(1) Page 1, line 3: the phrase “electropolymerized TiO2 nanotubes” causes misunderstanding, the TiO2 nanotube is fabricated by anodizing the Ti-6AL-4V alloy, similar issue occurs with Figure 7a.

(2) The use of electrochemical polymerization to deposit polymer electrolyte on the surface of electrode materials is very interesting. The electropolymerization mechanism should be discussed in the present work with a brief review of related literatures.

Author Response

Point 1: Page 1, line 3: the phrase “electropolymerized TiO2 nanotubes” causes misunderstanding, the TiO2 nanotube is fabricated by anodizing the Ti-6AL-4V alloy, similar issue occurs with Figure 7a.

Response 1: We agree with the reviewer. We decided to change “electropolymerized TiO2 nanotubes” with polymer-coated TiO2 NTs in the text to be consistent.

And the title using “electropolymerized TiO2 Nanotubes” has been changed.

Point 2: The use of electrochemical polymerization to deposit polymer electrolyte on the surface of electrode materials is very interesting. The electropolymerization mechanism should be discussed in the present work with a brief review of related literatures.

Response 2:  Thank you very much for this insightful comment. The electropolymerization mechanism has been added in the revised manuscript.

The reaction mechanism of MMA-PEG electropolymerization into TiO2 NTs is based on the formation of intermediate hydrogen-free radicals [37]. For cathodic applied potentials, the reduction of H+ producing H2 is accompanied by the formation of hydrogen-free radicals that can react with the monomers, leading to the polymerization of methyl methacrylate (PMMA). Figure 3 shows the cyclic voltammogram of the polymer-coated TiO2 NTs for 25 cycles. Clearly, the absolute value of the cathodic current at -1 V vs. Ag/AgCl (3M KCl) drops as the increment of cycles. The reason for the obvious fading can be explained by the successive growth of thin insulating polymer layers on the walls of TiO2 NTs.

Again, we appreciate all of your insightful comments. Thank you for taking the time and energy to help us improve the paper.